

# A preliminary study of carbon dioxide and methane emissions from patchy tropical seagrass meadows in Thailand

Muhammad Halim[1], Milica Stankovic[2,3] and Anchana Prathep[1,2]

[1] Division of Biological Science, Faculty of Science, Prince of Songkla University, Hat Yai, Songkhla, Thailand
[2] Excellence Center for Biodiversity of Peninsular Thailand, Faculty of Science, Prince of Songkla University, Hat Yai, Songkhla, Thailand
[3] Dugong and Seagrass Research Station, Prince of Songkla University, Hat Yai, Songkhla, Thailand

## ABSTRACT

**Background**: Seagrass meadows are a significant blue carbon sink due to their ability to store large amounts of carbon within sediment. However, the knowledge of global greenhouse gas (GHG) emissions from seagrass meadows is limited, especially from meadows in the tropical region. Therefore, in this study, $CO_2$ and $CH_4$ emissions and carbon metabolism were studied at a tropical seagrass meadow under various conditions.

**Methods**: $CO_2$ and $CH_4$ emissions and carbon metabolism were measured using benthic chambers deployed for 18 h at Koh Mook, off the southwest coast of Thailand. The samples were collected from areas of patchy *Enhalus acoroides*, *Thalassia hemprichii*, and bare sand three times within 18 h periods of incubation: at low tide at 6 pm (t0), at low tide at 6 am (t1), and at high tide at noon (t2).

**Results**: Seagrass meadows at Koh Mook exhibited varying $CO_2$ and $CH_4$ emissions across different sampling areas. $CO_2$ emissions were higher in patchy *E. acoroides* compared to patchy *T. hemprichii* and bare sand areas. $CH_4$ emissions were only detected in vegetated areas (patchy *E. acoroides* and *T. hemprichii*) and were absent in bare sand. Furthermore, there were no significant differences in net community production across sampling areas, although seagrass meadows were generally considered autotrophic. Koh Mook seagrass meadows contribute only slightly to GHG emissions. The results suggested that the low GHG emissions from Koh Mook seagrass meadows do not outweigh their role as significant carbon sinks, with a value 320 t $CO_{2-eq}$. This study provided baseline information for estimating GHG emissions in seagrass meadows in Thailand.

# INTRODUCTION

Blue carbon ecosystems, including seagrass meadows, make nature-based contributions to climate change mitigation by sequestering significant amounts of carbon and storing it within biomass and sediment over a long period of time (*Marbà et al., 2015*; *Serrano et al.,*

Corresponding author
Milica Stankovic, milica.s@psu.ac.th

2018). Globally, seagrass ecosystems contribute the equivalent of 10 to 15% of total carbon sequestration in the oceans (*Tang et al., 2018*). The contribution of seagrass to blue carbon storage is an effective long-term nature-based strategy against climate change (*Duarte, Sintes & Marbà, 2013*; *Kindeberg et al., 2018*; *Macreadie et al., 2021*; *Mtwana Nordlund et al., 2016*; *Rudd, 2014*).

However, seagrass ecosystems are also a source of carbon dioxide, methane and nitrous oxide emissions. The disturbance of seagrass meadows is the main driver of carbon release into the atmosphere (*Fourqurean et al., 2012*; *Pendleton et al., 2012*; *Thomson et al., 2019*). When seagrass meadows are disturbed, stored organic carbon becomes susceptible to destabilization by erosion, and organic carbon in sediment is exposed to oxygen, leading to increased remineralization (*Arias-Ortiz et al., 2018*; *Macreadie et al., 2014*; *Marbà et al., 2015*; *Pendleton et al., 2012*). Eventually, disturbances of organic carbon stocks in seagrass release $CO_2$ and other gases (*i.e.*, methane-$CH_4$ and nitrous oxide-$N_2O$) into the atmosphere (*Crooks et al., 2011*; *Rosentreter et al., 2021*). Seagrass degradation is estimated to release 0.15 Pg $CO_2$ per year, resulting from a seagrass conversion rate of 1.5% per year over a 30 Mha area globally (*Pendleton et al., 2012*). Furthermore, global potential $CH_4$ and $N_2O$ emissions from the seagrass meadows range from 0.09 to 2.7 Tg $yr^{-1}$ (*Garcias-Bonet & Duarte, 2017*) and 1.0 kg N $ha^{-1}$ $yr^{-1}$ (*Christensen & Rousk, 2024*), respectively. Seagrass net loss is 5,602 $km^2$ globally (survey area = 29,293 $km^2$) between 1880–2016 and 96 $km^2$ in the Indo-Pacific region (survey area = 592 $km^2$) between 1945–2016 (*Dunic et al., 2021*). In the long term, the potential $CO_2$ emissions related to seagrass loss approximately 23 million Mg $CO_2$ by 2100 (*Moritsch et al., 2021*).

Moreover, carbon metabolism in seagrass meadows is an important part of seagrass air-sea carbon fluxes. Seagrass carbon metabolism refers to the process of absorbing, utilizing and releasing carbon, including through photosynthesis and respiration (*Egea et al., 2020*). This process has to be considered when assessing carbon balance in seagrass meadows (*Ferguson et al., 2017*) where a positive carbon balance needs to be established by increasing carbon storage (*Jiménez-Ramos et al., 2023*). Disturbances such as eutrophication, thermal stress, storms, and coastal development activities (*Dahl et al., 2023*; *Salinas et al., 2020*) directly induce changes in the natural balance of the ecosystem, thereby reducing the capacity of seagrass as carbon store (*Egea et al., 2019a*).

In addition, within seagrass ecosystems, low levels of $CH_4$ are produced (*Al-haj, Chidsey & Fulweiler, 2022*; *Lyimo et al., 2018*). $CH_4$ production in seagrass meadows is controlled by environmental factors (*Qu et al., 2024*). High salinity, for instance, keeps $CH_4$ levels low (*Qu et al., 2024*) by increasing sulfate levels. Increased sulfate levels encourage the sulfate reduction process, which inhibits $CH_4$ production (*Asplund et al., 2022*). However, in seagrass ecosystems, $CH_4$ emissions can increase under certain conditions. For instance, excess nutrients are known to create ideal conditions for methanogenesis (*Al-haj, Chidsey & Fulweiler, 2022*; *Al-Haj & Fulweiler, 2020*; *Garcias-Bonet & Duarte, 2017*; *Rosentreter et al., 2021*), but further research on $CH_4$ dynamics in seagrass is required (*Oreska et al., 2020*; *Pendleton et al., 2012*).

Despite understanding GHG dynamics, global knowledge of $CO_2$ and $CH_4$ emissions, particularly natural $CO_2$ and $CH_4$ emissions in tropical seagrass, is still limited, especially

in Southeast Asia meadows (*Stankovic et al., 2023*). To fill this knowledge gap, this study estimated variations in $CO_2$ and $CH_4$ emissions to the atmosphere and carbon metabolism from tropical seagrass meadows in Thailand. The areas sampled included patchy meadows of *Enhalus acoroides* (Linnaeus f.) Royle, covering approximately 0.15 ha, and *Thalassia hemprichii* (Ehrenberg) Ascherson, covering approximately 0.10 ha, with bare sand as the dominant (>1 ha). This study provides baseline information for estimating greenhouse gas (GHG) emissions and for developing conservation strategies to mitigate GHG emissions in seagrass meadows in Thailand.

## MATERIALS AND METHODS

### Study area

Sampling was conducted Koh Mook, an island off the south-west coast of Thailand. The island is part of the Haad Chao Mai National Park in the Andaman Sea (Fig. 1A). Seven seagrass species have been recorded at Koh Mook, including *E. acoroides*, *T. hemprichii*, *C. rotundata*, *C. serrulata*, *H.ovalis*, *H. uninervis* and *S. isoetifolium* (*Nakaoka & Supanwanid, 2000*). Three species are dominant (*E. acoroides*, *T. hemprichii*, and *H. ovalis*) (*Nakaoka & Supanwanid, 2000*). The intertidal flat at Koh Mook is approximately 400 m from the shoreline (*Nakaoka & Supanwanid, 2000*), where the majority of the above species can be found. Samples were collected from areas of patchy *E. acoroides*, patchy *T. hemprichii*, and bare sand in the same seagrass meadow. Three benthic chambers were deployed at each area (Figs. 1B and 1C), for a total of nine chambers (Fig. 1B). The data were collected in June 2023, which is the beginning of the rainy season in Thailand (*Thai Meteorological Department, 2023*). Water temperatures during data collection ranged from 24.2° to 33.6°C while the average light intensity during the day was 94.93 µmol m$^{-2}$ s$^{-1}$.

### Design of benthic chamber, experimental setup and data collection

The transparent PVC benthic chambers were 120 cm in length for *E. acoroides* sampling, and 61.5 cm for *T. hemprichii* and bare sand sampling. They contained an air pocket and modified sampling ports for collecting gas and water samples (Fig. 2A).

Benthic chambers were installed approximately 15 cm into sediment (Fig. 2A). Incubation was conducted for 18 h, and samples were collected after 10 min of chamber set-up for the first sample collection at t0, 12 h after (t1) and 18 h after (t2). Approximately 5 mL of gas sample and 20–25 mL of water sample were collected from each chamber, using a syringe inserted into the sampling port (Fig. 2B). Samples were transferred to modified blood tube that were sealed with silicone to prevent sample leakage. The tubes were stored upside-down in a cold box/refrigerator (4 °C) until analysis in the lab (*Asplund et al., 2022*). Temperature and light intensity at each sampling location were recorded with a HOBO logger during the incubation time.

### $CO_2$ and $CH_4$ emissions, carbon metabolism with measuring changes in dissolved oxygen concentration

$CO_2$ and $CH_4$ emissions from seagrass to the atmosphere were estimated following the method of *Asplund et al. (2022)*. In the laboratory, samples were analyzed by using a

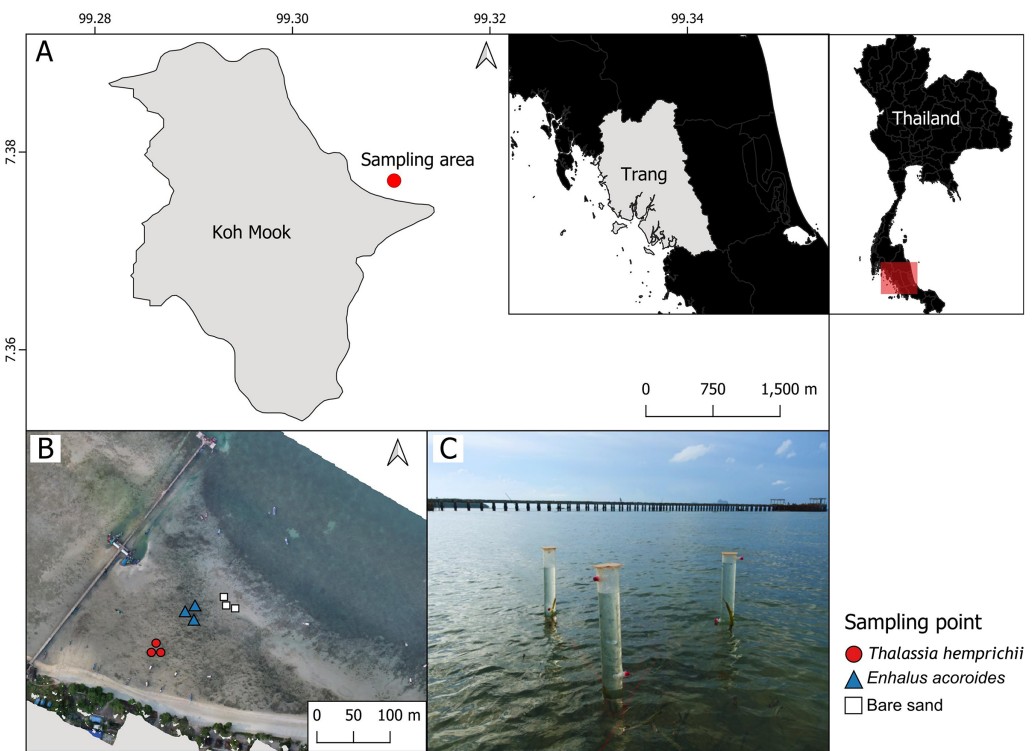

**Figure 1 The study sites in the intertidal zone at Koh Mook.** The study site in the intertidal zone at Koh Mook (A) included areas of patchy *E. acoroides*, areas of patchy *T. hemprichii* and bare sand areas (B). Benthic chambers were installed at three points in each of the three areas (C). Notes: The map was created using the free and open source QGIS (A), and the drone image of the seagrass was obtained from Rattanachot et al. 2024 (in preparation) (B). Photo credit: taken by SSRU Lab Member (C).

method described in *Kitpakornsanti, Pengthamkeerati & Limsakul (2022)*. A gas chromatograph (Agilent 6890N GC, Agilent, Santa Clara, CA, USA) equipped with an electron capture detector was used to determine $CO_2$ and $CH_4$ production from sediment. Quality control was checked using 1,000 rpm $CO_2$ and $CH_4$ on each sample, and standard deviation of replicates was less than 5%. The $CO_2$ and $CH_4$ emissions from seagrass sediment were calculated using the following equation (Eq. (1)) from *Bulmer et al. (2017)*,

$$F = \left(\frac{dG}{dt}\right) \times V \times P/(R \times T \times A) \tag{1}$$

where **F** is the $CO_2$ and $CH_4$ emissions (mmol m$^{-2}$ h$^{-1}$); **dG/dt** is the slope of the linear regression between gas concentrations and deployment time (ppmv h$^{-1}$); **V** is the headspace volume (L) of the chamber; **P** is the barometric pressure (1 atm); **R** is the ideal gas constant ($8.205746 \times 10^{-5}$ atm m$^2$ K$^{-1}$ mol$^{-1}$); **T** is the average temperature (°K) inside the closed chamber at each measurement in the field; **A** is the sectional area (m$^2$) of the bottom of the closed chamber.

Furthermore, community carbon metabolism was estimated from the dissolved oxygen (DO) concentration (*Jiménez-Ramos et al., 2022*) determined using the Winkler titration method. The collected water samples were kept in a cold box/refrigerator (4 °C) until they

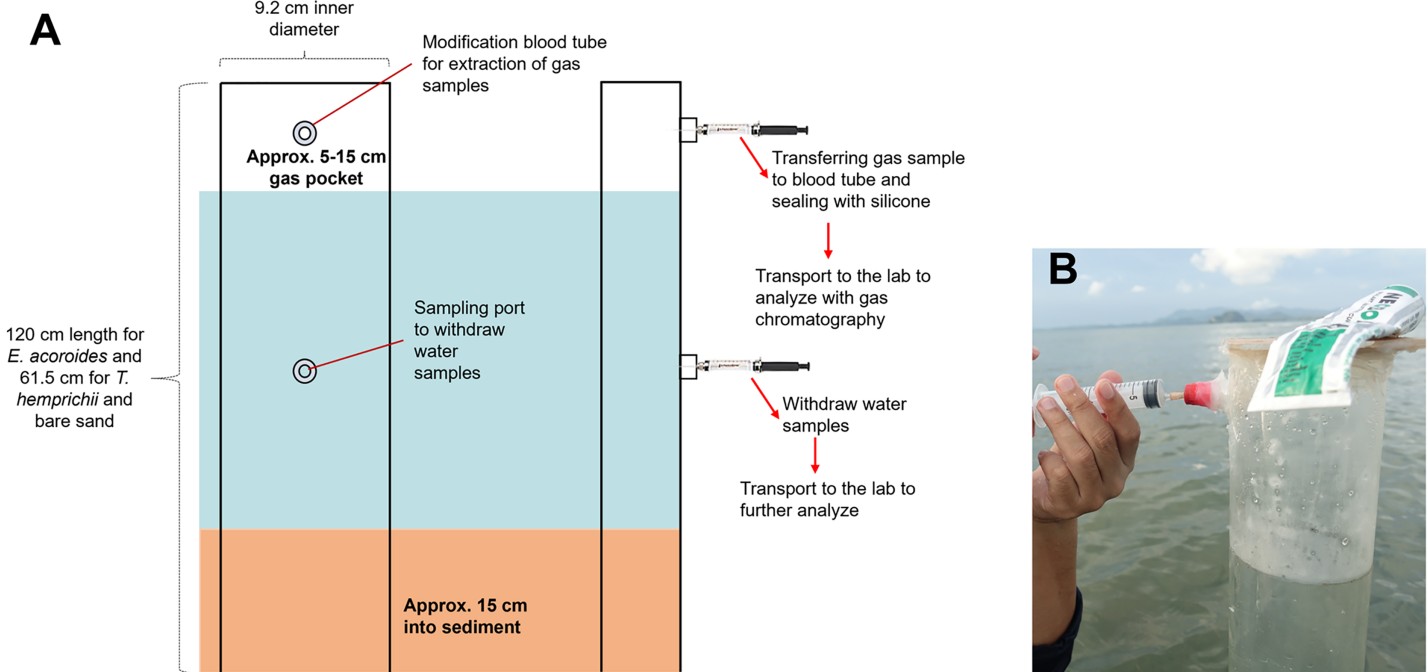

**Figure 2 Benthic chamber method used for the sample collection.** Benthic chamber method used for the sample collection in the field (A); the process of withdrawing samples from the chamber using a syringe (B). Photo credit: Designed by Muhammad Halim (A) and taken by SSRU Lab Member (B).

were analyzed in the laboratory (*Egea et al., 2019b*). Hourly community respiration rates (CR) were estimated as the difference in DO concentration between t1 and t0 samples divided by the time elapsed between sampling, using the following equation (Eq. (2)) (*Jiménez-Ramos et al., 2022*):

$$CR\left(\frac{mmolO_2}{m^2d}\right) = \frac{DO_{t1}\left(\frac{mgO_2}{1}\right) - DO_{t0}\left(\frac{mgO_2}{1}\right)}{\Delta T_{t0} - T_{t1}\ (h)} \times \frac{volume\ (l)}{Area\ (m^2)} \times \frac{1\ mmolO_2}{32\ mgO_2} \quad (2)$$

where $DO_{t1}$ and $DO_{t0}$ are the DO concentrations at t1 and t0, $\Delta T$ is the time elapsed between sampling events, and volume and area are for benthic chambers.

Hourly rates of net community production (NCP) were estimated as the difference in DO concentration between t2 and t1 samples divided by the time elapsed between sampling using the following equation (Eq. (3)):

$$NCP\left(\frac{mmolO_2}{m^2d}\right) = \frac{DO_{t2}\left(\frac{mgO_2}{1}\right) - DO_{t1}\left(\frac{mgO_2}{1}\right)}{\Delta T_{t1} - T_{t2}\ (h)} \times \frac{volume\ (l)}{Area\ (m^2)} \times \frac{1\ mmolO_2}{32\ mgO_2} \quad (3)$$

where $DO_{t2}$ and $DO_{t1}$ are the DO concentrations at t2 and t1, $\Delta T$ is the time elapsed between sampling events, and volume and area are for benthic chambers.

Hourly rates of community gross primary production (GPP) were calculated as the sum of hourly rates of CR and NCP:

$$GPP = CR + NCP. \tag{4}$$

Finally, daily rates of community GPP, CR, and NCP were estimated following from:

$$GPP = GPP \times photoperiod \; (h) \tag{5}$$

$$CR = CR \times 24 \; h \tag{6}$$

$$NCP = GPP - CR \tag{7}$$

where photoperiod corresponds to the number of sunlight hours measured on each sampling day. Metabolic rates in DO units were converted to carbon units assuming photosynthetic and respiratory quotients of 1 (values are used widely in seagrass studies) (*Egea et al., 2019b*).

## Statistical analysis

Data were checked for normality and transformed as necessary before further analysis to meet the assumptions of the ANOVA. Data estimates throughout this work were presented as averages with the standard deviation (SD). ANOVA was used to test significant differences between sampling areas (bare sand, *E. acoroides* and *T. hemprichii*) and $CO_2$ emissions, and between sampling times (t0, t1 and t2) and $CO_2$ emissions. A *post-hoc* test was performed to compare differences in $CO_2$ emissions between the analyzed groups, including sampling times across all three sampling areas. Furthermore, MANOVA was used to test significant differences between carbon metabolism with both sampling area and sampling time. All statistical analyses were done in R statistical software version 4.3.1 (*R Studio: Integrated Development for R, 2021*), using the following packages: tidyverse 2.0.0 (*Wickham et al., 2019*), agricolae 1.3-6 (*de Mendiburu, 2023*), emmeans 1.8.8 (*Lenth, 2023*), multcomp 1.4-25 (*Hothorn, Bretz & Westfall, 2008*), car 3.1-2 (*Fox & Weisberg, 2019*), ggplot2 3.4.3 (*Wickham, 2016*), MASS 7.3-60 (*Venables & Ripley, 2002*), mvoutlier 2.1.1 (*Filzmoser & Gschwandtner, 2021*), pastecs 1.3.21 (*Ibanez, 2018*), mvnormtest 0.1-9 (*Jarek, 2012*), and reshape2 1.4.4 (*Wickham, 2007*).

## RESULTS

### Carbon dioxide emissions from Koh Mook seagrass meadows

ANOVA indicated a significant difference ($p < 0.001$) between $CO_2$ emissions and sampling area, where the average $CO_2$ emissions were significantly higher ($p < 0.05$) in the *E. acoroides* area ($0.143 \pm 0.000788 \; \mu mol \; CO_2 \; m^{-2} \; h^{-1}$) than in the *T. hemprichii* area ($0.0489 \pm 0.000403 \; \mu mol \; CO_2 \; m^{-2} \; h^{-1}$). Meanwhile, in the bare sand area, the average $CO_2$ emissions were significantly the lowest ($p < 0.05$) at $0.0082 \pm 0.0000404 \; \mu mol \; CO_2 \; m^{-2} \; h^{-1}$ (Fig. 3).

Furthermore, the *post-hoc* test showed that $CO_2$ emissions were not significantly different ($p > 0.05$) between sampling times across all three sampling areas (Fig. 4). However, in the *E. acoroides* area, the average $CO_2$ emissions were slightly higher at t1 ($0.1440 \pm 0.00080 \; \mu mol \; CO_2 \; m^{-2} \; h^{-1}$) than at t0 and t2 ($0.1424 \pm 0.00080$ and $0.1431 \pm$
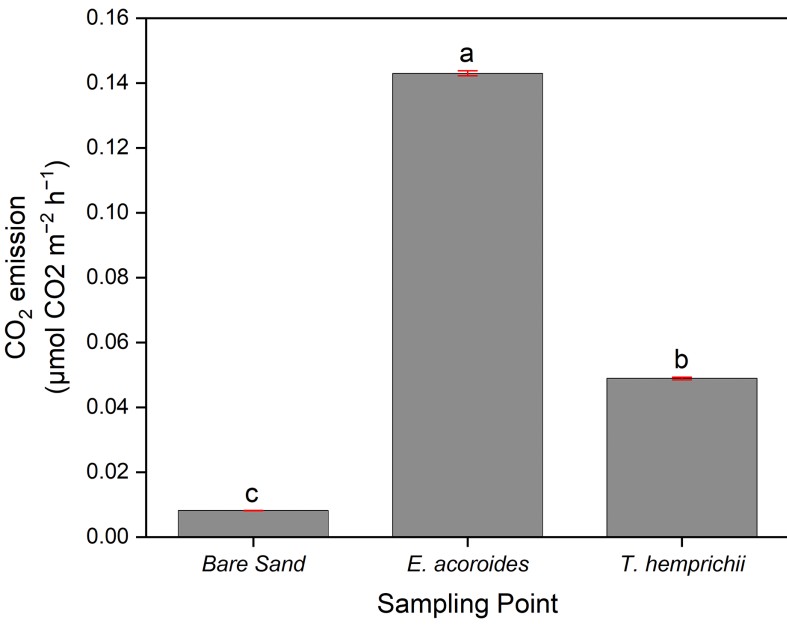

**Figure 3 The average CO₂ sediment-water surface emissions from seagrass patches and bare sand at Koh Mook.** The average CO₂ sediment–water surface emissions from seagrass patches and bare sand at Koh Muk recorded during 18 h of benthic chamber incubation.

0.00080 $\mu$mol $CO_2$ $m^{-2}$ $h^{-1}$, respectively). The same pattern was found in the bare sand area, where $CO_2$ emissions were also slightly higher at t1 ($0.00825 \pm 0.00042$ $\mu$mol $CO_2$ $m^{-2}$ $h^{-1}$) than t0 ($0.00816 \pm 0.00042$ $\mu$mol $CO_2$ $m^{-2}$ $h^{-1}$) and t2 ($0.00819 \pm 0.00042$ $\mu$mol $CO_2$ $m^{-2}$ $h^{-1}$). In the *T. hemprichii* area, $CO_2$ emissions were only recorded at t0 ($0.0485 \pm 0.02817$ $\mu$mol $CO_2$ $m^{-2}$ $h^{-1}$) and t1 ($0.0491 \pm 0.02817$ $CO_2$ $m^{-2}$ $h^{-1}$) as the benthic chambers were slightly damaged at t2, and $CO_2$ emissions could not be measured.

### Seagrass community carbon metabolism

Overall, the MANOVA showed that there was no significant difference ($p > 0.05$) in daily carbon metabolism across the different sampling areas (Fig. 5). However, some slight trends were observed in GPP, CR, and NCP, such that the values were always higher in the vegetated areas than in the bare sand area (Fig. 5).

The daily average GPP was highest in the *E. acoroides* area, followed by the *T. hemprichii* and bare sand areas, with average values of $1.61 \pm 3.17$, $1.53 \pm 0.62$, and $0.71 \pm 1.78$ mmol C $m^{-2}$ $d^{-1}$, respectively (Fig. 5A). Daily CR was in the *T. hemprichii* area at $-3.50 \pm 0.75$, followed by the *E. acoroides* area at $-3.33 \pm 2.62$ and bare sand area at $-2.30 \pm 1.38$ mmol C $m^{-2}$ $d^{-1}$ (Fig. 5B). Daily NCP was highest in the *T. hemprichii* area ($5.04 \pm 0.35$ mmol C $m^{-2}$ $d^{-1}$), slightly lower in the *E. acoroides* area ($4.95 \pm 1.01$ mmol C $m^{-2}$ $d^{-1}$) and lowest in the bare sand area ($3.01 \pm 0.41$ mmol C $m^{-2}$ $d^{-1}$) (Fig. 5C). As they are >1, these values of NCP suggest that the seagrass meadows at Koh Mook are autotrophic.

MANOVA showed no significant difference ($p > 0.05$) between seagrass community carbon metabolism, including GPP, CR, and NCP, and sampling time (t0, t1 and t2) (Fig. 6). However, some patterns were observed across sampling times and areas. We
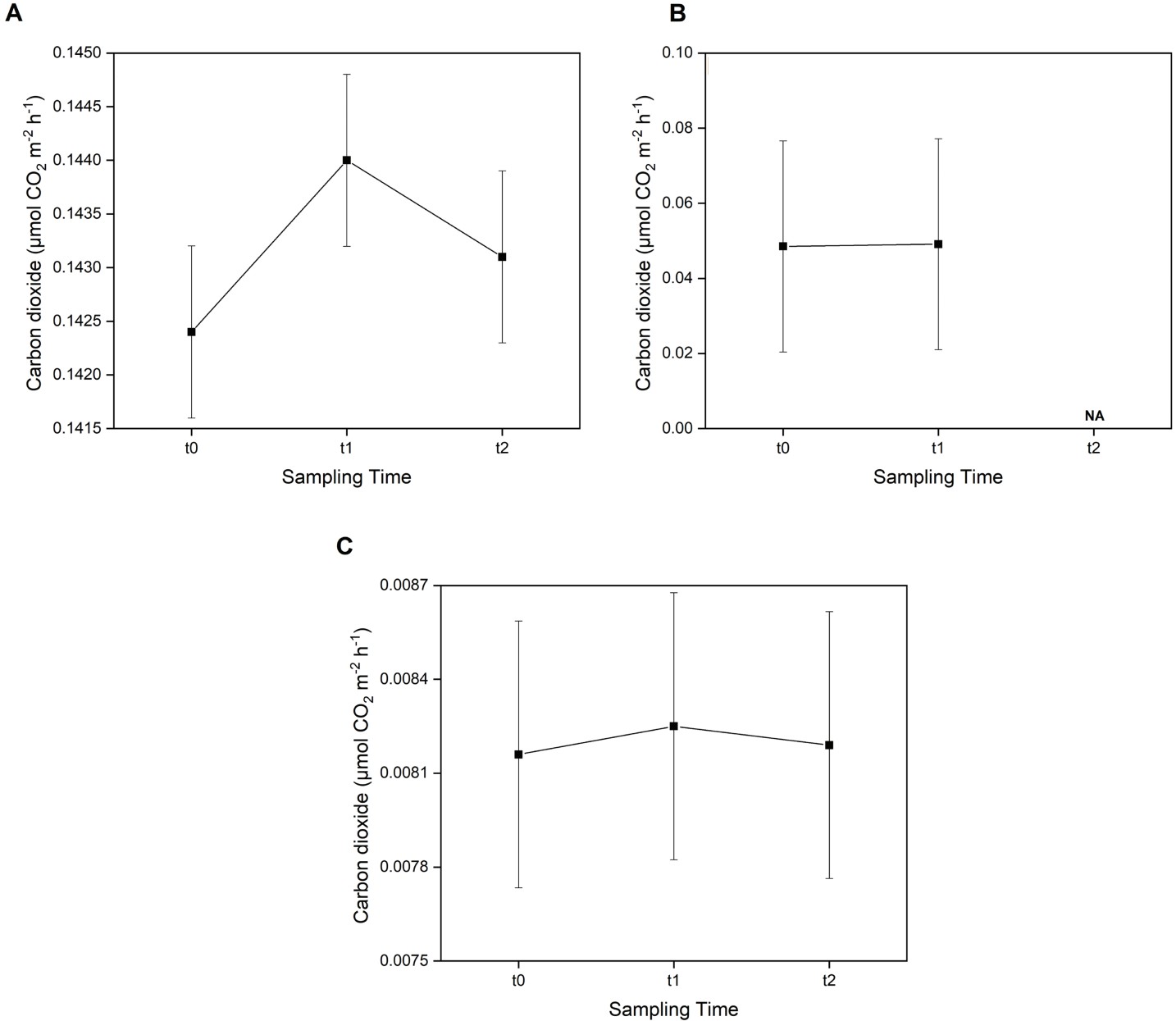

**Figure 4  Concentration of CO$_2$ emissions in two seagrass species and bare sand area.** Concentrations of CO$_2$ emissions in the *E. acoroides* area (A), *T. hemprichii* area (B), and bare sand area (C) at three different sampling times (t0: 6 pm on the first day; t1: 6 am on the second day; and t2: 12 noon on the second day). Note: NA = data not available due to damaged chambers.

found that the GPP and CR in the *T. hemprichii* and *E. acoroides* areas followed the same pattern (Figs. 6 A and 6B), with higher values at t1 (GPP: 1.575 and 2.007 mmol C m$^{-2}$ d$^{-1}$, CR: −2.836 and −1.556 mmol C m$^{-2}$ d$^{-1}$, respectively) than t0 (GPP: 1.499 and 1.223 mmol C m$^{-2}$ d$^{-1}$, CR: −4.171 and −5.116 mmol C m$^{-2}$ d$^{-1}$, respectively). However, there were some differences in GPP and CR between the two seagrass species. In the *E. acoroides* area, GPP was 0.78 mmol C higher at t1 than t0, and CR was 3.56 mmol C higher at t1 than t0. Meanwhile, in the *T. hemprichii* area, GPP was only 0.07 mmol C higher at t1 than t0, and
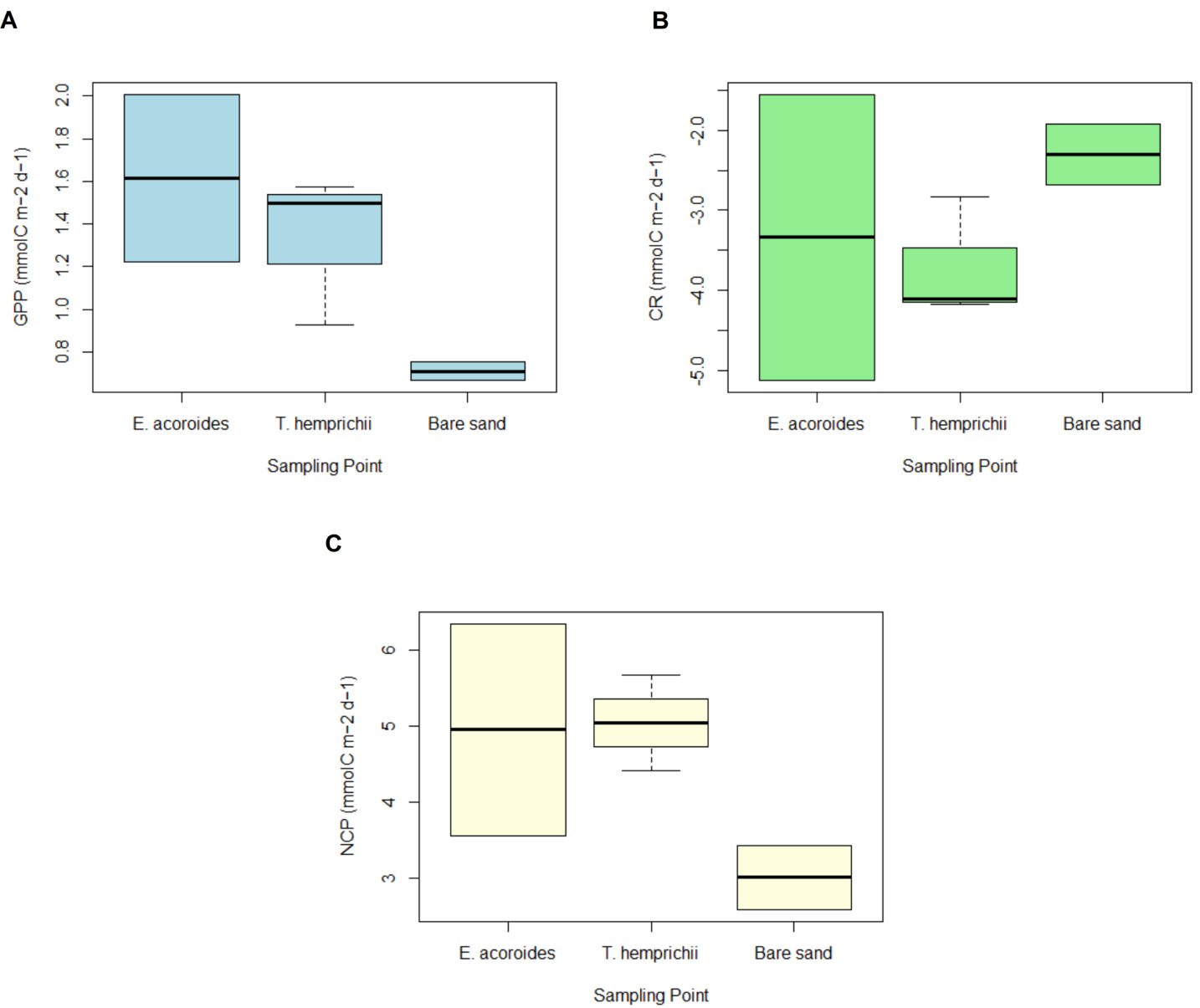

**Figure 5 GPP, CR and NCP of two seagrass species and bare sand area.** The average value ± SD of (A) daily gross primary production (GPP); (B) daily community respiration (CR); and (C) daily net community production (NCP) at various sampling areas at Koh Mook, Thailand during 18 h of benthic chamber incubation.

CR 0.91 mmol C higher (Figs. 6A and 6B). On the other hand, in the bare sand area, GPP was slightly higher at t0 (0.754 mmol C m$^{-2}$ d$^{-1}$) than t1 (0.668 mmol C m$^{-2}$ d$^{-1}$), while CR was slightly higher at t1 ($-$1.919 mmol C m$^{-2}$ d$^{-1}$) than t0 ($-$2.681 mmol C m$^{-2}$ d$^{-1}$) (Figs. 6A and 6B). Meanwhile, the NCP values at all sampling areas were higher at t0 (3.435, 5.671, and 6.340 mmol C m$^{-2}$ d$^{-1}$, respectively) than t1 (2.587, 4.411, and 3.564 mmol C m$^{-2}$ d$^{-1}$, respectively) (Fig. 6C).

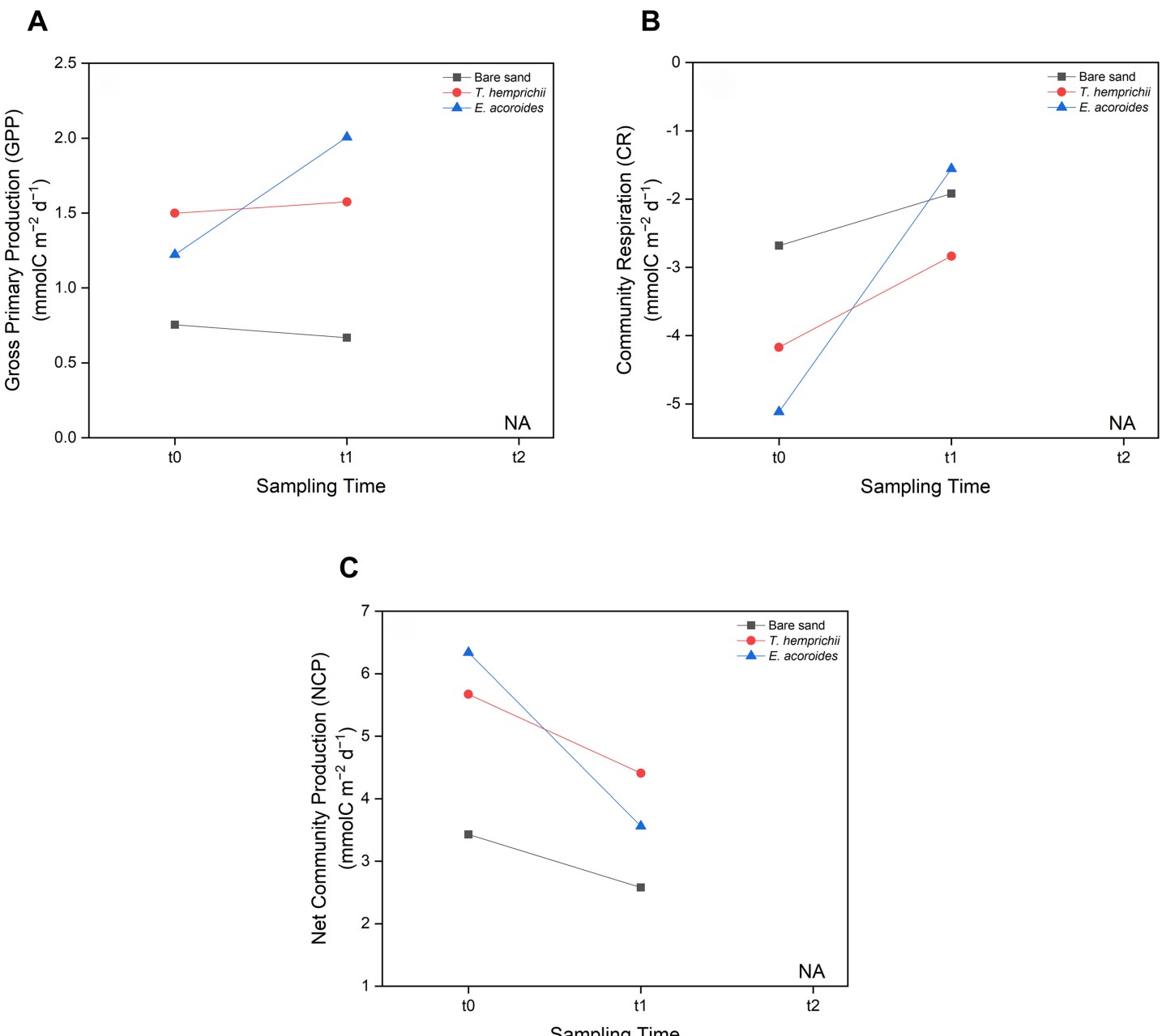

**Figure 6 Carbon metabolism of two seagrass species and bare sand area.** Seagrass carbon metabolism, including (A) Gross Primary Production, (B) Community Respiration and (C) Net Community Production at different sampling times (t0: 6 pm on the first day, t1: 6 am on the second day, and t2: 12 noon on the second day). Note: Data for t2 is not included in this research due to errors during water sampling and lab analysis.

## Methane emissions

$CH_4$ emissions to the atmosphere from the Koh Mook seagrass meadows were at low levels, only detected in samples from vegetated areas, where the values were higher in the *E. acoroides* area than the *T. hemprichii* area (Fig. 7A). Furthermore, $CH_4$ emissions were only detected at t0 (2.09 µg $CH_4$ m$^{-2}$ h$^{-1}$ in the *E. acoroides* area and 1.35 µg $CH_4$ m$^{-2}$ h$^{-1}$
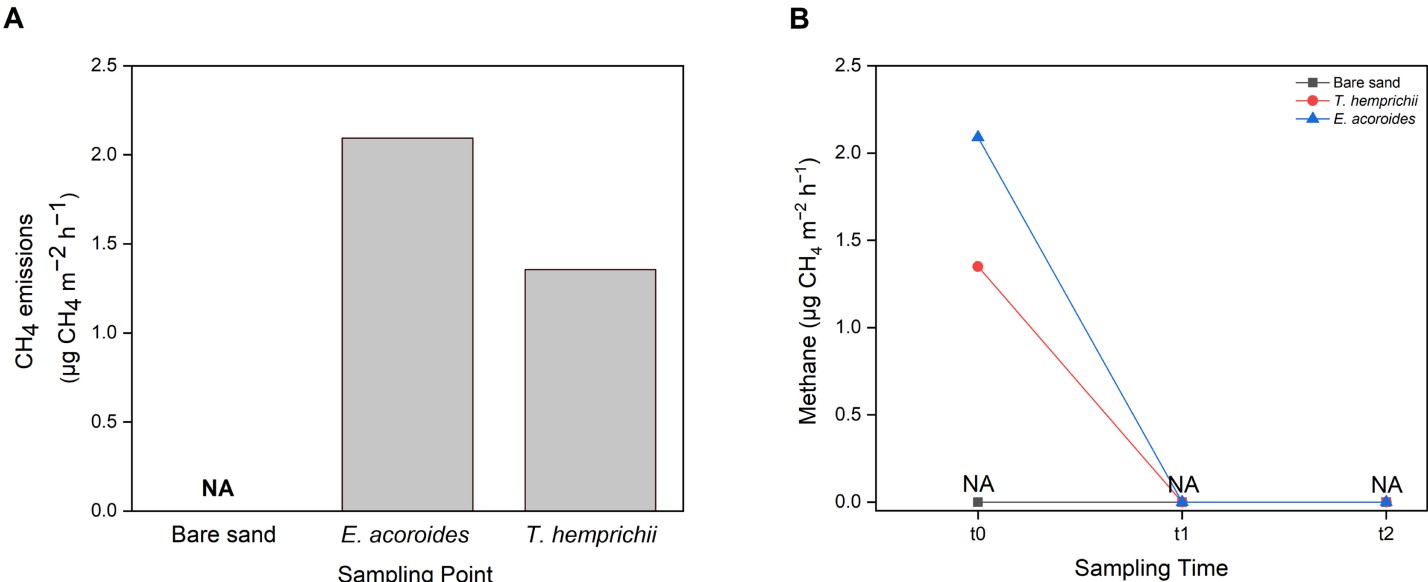

**Figure 7 Sediment-water CH$_4$ emissions of two different seagrass species and bare sand area.** Sediment–water surface CH$_4$ emissions from (A) different sampling areas; (B) at different sampling times. Note: NA indicates that CH$_4$ emissions were not detected.

in the *T. hemprichii* area) (Fig. 7B). Additionally, CH$_4$ emissions were undetected in the bare sand area (Figs. 7A and 7B).

## DISCUSSION

### Carbon dioxide emissions

The results showed that CO$_2$ emissions from seagrass meadows to the atmosphere in Koh Mook Island, Thailand, were relatively low (less than 10-fold) compared to emissions reported in several previous studies, but different methodologies were employed across these studies (*Burkholz, Garcias-bonet & Duarte, 2020*; *Roca et al., 2022*).

Although the meadows at Koh Mook release CO$_2$, the levels are low and do not outweigh their role in carbon sequestration. The low CO$_2$ emissions at Koh Mook are suppressed by the condition of the seagrass meadows, which are not under localized threats significant enough to change the sediment condition from anoxic to oxic (*Lovelock, Fourqurean & Morris, 2017*). Potential disturbances in this area include local community activities, such as monthly gleaning and small boats, which do not threaten the seagrass carbon stock. Thus, they will continue to contribute to carbon sequestration as long as local conditions remain the same (*Dahl et al., 2023*).

There were significant differences in CO$_2$ emissions between sampling areas. Higher values were found in vegetated areas compared to unvegetated areas, which was the pattern reported by *Latifah et al. (2023)* in mixed seagrass meadows with five species (*E. acoroides*, *T. hemprichii*, *C. rotundata*, *O. serrulata*, and *H. ovalis*) at Karimun Jawa, Indonesia and by *Burkholz, Garcias-bonet & Duarte (2020)* in *H. stipulacea* meadows in the Red Sea. We suspect that this pattern is due to the presence of seagrass, which store organic carbon and increase oxygen input into the sediment, thereby driving remineralization, which causes an

increase in the production of $CO_2$ (*Dahl et al., 2023*). *Banerjee et al. (2019)* proposed that $CO_2$ emissions were relatively higher in vegetated areas because the composition and action of the more active and dominant microbes drove GHG release. Furthermore, vegetated areas contribute to the production of particulate organic matter and trap terrigenous dissolved organic carbon (tDOC), facilitating the remineralization of particulate matter, which leads to $CO_2$ emissions from coastal waters (*Wahyudi et al., 2021*; *Zhou et al., 2021*). This process decreases the pH in shallow waters (seagrass meadows) and encourages the release of $CO_2$ into the atmosphere (*Capelle et al., 2020*; *Zhou et al., 2021*). Moreover, the photosynthesis that occurs only in vegetated areas consumes $CO_2$, driving the deposition of calcium carbonate ($CaCO_3$), which increases the carbonate ($CO_3^{2-}$) saturation of the water and increases water pH (*Buapet, Gullström & Björk, 2013*; *Latifah et al., 2023*; *Leclercq, Gattuso & Jaubert, 2000*; *Semesi, Beer & Björk, 2009*). This process increases aquatic $pCO_2$ until the pressure is high enough to release $CO_2$ into the atmosphere (*Kalokora et al., 2020*; *Latifah et al., 2023*; *Leclercq, Gattuso & Jaubert, 2000*).

We identified a slight difference in $CO_2$ emission values between sampling times with higher values at t1 (around 6-am on the second day). However, the statistical results did not show significant differences between $CO_2$ emissions and sampling time. The same $CO_2$ emission pattern was reported by *Ismail et al. (2023)* in mesocosm experiments on different plant community compositions, including seagrass (*T. hemprichii*) and calcifying algae (*Hydrolithon sp.*). The factors that cause small fluctuations in $CO_2$ emissions are photosynthesis and respiration. The slightly increased $CO_2$ emission at t1 was probably caused by nocturnal respiration from plant communities and sediments. The lower $CO_2$ emission values at t2 (around 12 noon on the second day) were the result of diurnal photosynthesis which absorbs $CO_2$ (*Ismail et al., 2023*). The lowest $CO_2$ emissions were recorded at t0 (around 6 pm on the first day) because the net $CO_2$ release decreased in the afternoon, as was found in *Ismail et al. (2023)*. Possibly due to DO production in the water by the photosynthesis process throughout the day encourages small $CO_2$ production in the late afternoon (*Ismail et al., 2023*).

## Seagrass community carbon metabolism

Carbon metabolism in seagrass meadows is typically intensive, supporting high GPP (*Hemminga & Duarte, 2000*). The present results indicate that the average daily GPP ranged from 0.711 to 1.33 mmol C $m^{-2}$ $d^{-1}$, with daily CR between −2.30 and −3.70 mmol C $m^{-2}$ $d^{-1}$. Consequently, we observed that daily NCP ranged from 3.01 to 5.04 mmol C $m^{-2}$ $d^{-1}$, which is within the range (0.6 to 6.8) reported for *Thallasia testudenum* meadows in Texas (*Ziegler & Benner, 1999*) but lower than reported for *Cymodocea nodosa* meadows (18 ± 4 mmol C $m^{-2}$ $d^{-1}$) in winter at Cadiz Bay, Spain (*Egea et al., 2020*). The negative CR value indicates that Koh Mook seagrass meadows absorb more $CO_2$ than they release, promoting balance in the seagrass community carbon metabolism and increasing NCP. These results indicate that the seagrass meadows at Koh Mook generally exhibit autotrophic conditions (NCP > 1) or contribute to a carbon sink. This preliminary research suggests that these meadows have the potential to accumulate organic carbon.

Seagrass carbon metabolism rates showed variations between the vegetated areas (patchy *E. acoroides* and *T. hemprichii*) and unvegetated bare sand. Although GPP, CR and NCP were not significantly different between sampling areas and sampling times, the presence of patchy *E. acoroides* and *T. hemprichii* influenced carbon metabolism at the study site. Higher carbon metabolism can be driven by seagrass biomass, as it provides a complex habitat that directly impacts carbon metabolism (*Barrón et al., 2004*; *Egea et al., 2020*; *Johnson et al., 2020*). Furthermore, the above-ground biomass of *E. acoroides* and *T. hemprichii* at Koh Mook contributed to carbon metabolism by providing photosynthetic tissue (*Jiménez-Ramos et al., 2022*).

Seagrass carbon metabolism, especially NCP, was higher at t0 (around 6 pm on the first day) than t1 (around 6 am on the second day). This pattern was possibly influenced by the short incubation time, as NCP tends to decrease with prolonged incubation (*Olivé et al., 2016*), which reduces oxygen levels in the benthic chamber, thereby reducing the NCP budget. Moreover, the period of darkness leading up to t1 increased CR, which utilized $O_2$ and released $CO_2$ (*Olivé et al., 2016*).

## Methane emissions

Seagrass meadows are also considered a natural source of small amounts of $CH_4$ (*Asplund et al., 2022*). At the study site, sediment–water methane was only detected at t0 at the *E. acoroides* and *T. hemprichii* areas. $CH_4$ was not detected in the unvegetated area or at t1 and t2, most likely because the sampling method used was not sensitive enough to capture the low methane levels released at those sampling times, mainly when sampling is conducted at a specific time when methane production is low. The $CH_4$ values recorded in this study are within the range reported for *Zostera marina* meadows in Nordic waters (0.3–30 µg $CH_4$ m$^{-2}$ h$^{-1}$) (*Asplund et al., 2022*). Our results suggest that the seagrass meadows at Koh Mook contribute only slightly to $CH_4$ emissions. The low $CH_4$ production at the Koh Mook meadow might be related to the salinity of these waters. Higher salinity allows higher sulfate levels to develop in seawater, thus microbes that utilize sulfate become more dominant, and methanogenesis is inhibited (*Schorn et al., 2022*).

Even though $CH_4$ values were low, this gas is nonetheless a GHG with a Sustained-Flux Global Warming Potential (SGWP) 45–96 times greater than $CO_2$ (*Neubauer & Megonigal, 2015*). Therefore, small emissions of $CH_4$ can still offset the carbon sink in seagrass ecosystems and must be taken into consideration. For instance, $CH_4$ emissions from *P. oceanica* meadows could offset less than 1% of the carbon buried in sediment (*Yau et al., 2022*), while in mangrove ecosystems, $CH_4$ emissions could offset up to 18% of carbon burial (*Rosentreter et al., 2018*). Although $CH_4$ emissions were only detected in the vegetated areas at Koh Mook, the contribution of seagrass to $CH_4$ emissions should not be ignored since seagrass meadows can facilitate $CH_4$ production. Furthermore, under certain conditions, such as increasing temperature (*Begum et al., 2020*), low salinity (*Schorn et al., 2022*) and high organic carbon content (*Asplund et al., 2022*), $CH_4$ production can disrupt the role of seagrass as a GHG sink. According to *Schorn et al. (2022)*, the release of $CH_4$ by seagrass will offset carbon burial by 4 to 5%. Finally, $CH_4$ emissions were only found at t0

after 10 min of chamber setup, and it could be that sediment disturbance during chamber setup released $CH_4$ into the water column.

### Research limitations and improvements

This preliminary study has several limitations that need to be considered, including the duration of $CO_2$ and $CH_4$ emission measurements, which was only 18 h, thus not fully representing the daily emission values for 24 h.

We emphasize the importance of improving this study in the future by assessing long-term trends and exploring different environmental conditions, such as temperature and salinity, to ensure a more comprehensive understanding of the contribution of seagrass meadows in Thailand to GHG emissions. The benthic chamber used in our research proved to be a more practical, effective, and reliable method of measuring GHG emissions in remote areas, such as Koh Mook, than methods such as eddy-covariance (*Tokoro et al., 2007*). However, this method also has weaknesses because it can increase surface gas flux due to the pressure difference between water and air (*Ismail et al., 2023*).

In future research, several improvements and considerations must be made to ensure the validity of capturing GHG emissions from seagrass meadows with a benthic chamber. Local tides, waves, substrates, and coverage are important factors, and gas leakage from the chamber must be addressed. Additionally, this preliminary study had limited spatial sampling coverage and the number of benthic chambers. In the future, the number of chambers will increase to ensure that emissions measurements fully represent the scale of the large study area. Moreover, for a comprehensive understanding of the carbon dynamics in seagrass, it is also recommended that water- and air-$pCO_2$ be measured. Furthermore, for a comprehensive understanding of the carbon dynamics in seagrass, it is also recommended that water- and air-$pCO_2$ be measured.

## CONCLUSIONS

Seagrass meadows at Koh Mook in Thailand have relatively low $CO_2$ emissions compared to global values, suggesting that these meadows are effective for long-term carbon storage. Additionally, the meadows exhibit autotrophic conditions and have the potential to accumulate organic carbon. $CH_4$ emissions from the meadows are low but still need consideration due to their potential as a greenhouse gas. However, this study has limitations, and further research is needed to fully understand the contribution made by seagrass meadows in Thailand to GHG emissions.

## ACKNOWLEDGEMENTS

The authors thank the Seaweed and Seagrass and Research Unit (SSRU) members for their assistance in the field and laboratory work.

### Funding

This study was supported by a PSU-Ph.D scholarship (No. PSU_PHD2564-05) from Prince of Songkla University, PSU-TUYF Charitable Trust Fund, and Program Management Unit for Human Resources & Institutional Development, Research and Innovation (No. B13F660071). The funders had no role in study design, data collection and analysis, decision to publish, or preparation of the manuscript.

### Grant Disclosures

The following grant information was disclosed by the authors:
Prince of Songkla University: PSU_PHD2564-05.
PSU-TUYF Charitable Trust Fund.
Program Management Unit for Human Resources & Institutional Development, Research and Innovation: B13F660071.

### Competing Interests

The authors declare that they have no competing interests.

### Author Contributions

- Muhammad Halim conceived and designed the experiments, performed the experiments, analyzed the data, prepared figures and/or tables, authored or reviewed drafts of the article, and approved the final draft.
- Milica Stankovic conceived and designed the experiments, prepared figures and/or tables, authored or reviewed drafts of the article, and approved the final draft.
- Anchana Prathep conceived and designed the experiments, authored or reviewed drafts of the article, and approved the final draft.

### Data Availability

   The raw data is available in the Supplemental File.

### Supplemental Information

Supplemental information for this article can be found online at http://dx.doi.org/10.7717/peerj.18087#supplemental-information.

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
