# Peer review of "A preliminary study of carbon dioxide and methane emissions from patchy tropical seagrass meadows in Thailand"

_PeerJ, doi:10.7717/peerj.18087_

## Round 0.1 · original submission · Major Revisions

The authors need to revise well based on 3 reviewers' comments. One reviewer advised to reject based on the experiment time and literature review, in addition to English language, so authors need to follow those comments seriously and resubmit. English needs to be revised well.

Reviewer 1 ·

Basic reporting

Overall, the manuscript presents valuable research findings on greenhouse gas emissions and carbon metabolism in tropical seagrass meadows, with potential implications for ecosystem management and climate change mitigation strategies. The findings contribute valuable insights into CO2 and CH4 fluxes and carbon metabolism in seagrass meadows, providing baseline information for estimating greenhouse gas emissions in similar ecosystems in Thailand and beyond. However, there are several things to note.

Experimental design

1.Throughout the manuscript, the authors use the term carbon emissions interchangeably with carbon flux (air-sea CO2 flux). The authors need to clarify whether they mean carbon emissions to the atmosphere or to the water column or both; since the term will affect the context.
2.In the introduction, it was stated that seagrass ecosystems can emit carbon in the form of CO2 and CH4. The author emphasises that this is particularly the case because it is caused by disturbance to the ecosystem. However, the author then states that the research was conducted to look at natural emissions in seagrass ecosystems. In this case, I think the author needs to straightforwardly convey the importance of assessing natural CO2 and CH4 emissions in seagrass ecosystems.
3.Line 90-91: “The intertidal zone at Koh Mook is approximately 400 m wide” >> I would expect the zone is in meter sequare. Do you mean 400 m2?
4.Line 93-97: The deployment of only three benthic chambers per area seems inadequate to capture the variability of benthic conditions in the given regions. With such a small number, the findings may lack robustness and generalizability. Additionally, it's unclear how the placement of these chambers was determined and whether it adequately represents the entire area under study.
Line 93-97: The beginning of a rainy season could introduce significant variability in environmental conditions compared to other times of the year, potentially confounding the interpretation of the results. The author may explain the reason of choosing the experiment periods.

Validity of the findings

6.Line 96-97: The range of water temperatures mentioned (24.2 to 33.6°C) is quite broad. Such variability could significantly influence the benthic ecosystem and might require further investigation or explanation regarding its impact on the observed phenomena. Is there any effect of sampling periods? Or sampling time? As T is part of CO2 or CH4 flux equation, the broad range of temperature may affect the broad flux variability.
7.Line 225-230: the author mention of Research by Roca et al 2022 and Latifah et al 2023. Roca et al. Conducted research to investigate the degraded seagrass, therefore it is expected that the CO2 flux would be higher compared your study. Latifah et al. Actually studied the CO2 flux in term of differences of pCO2 water and air, which is by method is different with your experiment. The difference then raise a question, why did the author use the mesocosm chamber for the experiment? Since the mesocosm experiment may only capture micro-scale conditions.
Line 240-243: carbon (gas) flux in the coastal ecosystem and estuary is caused by a complex process including the remineralization of terrestrial dissolved organic carbon to the dissolved inorganic carbon and CO2 (see Zhou et al 2021). In addition, within vegetated seagrass ecosystem, the particulate matter may have a longer residence time in the water column (because trapped by the seagrass vegetation) compared to bared sand area. The consequence is that the particulate matter may have longer time for decomposition process, which lead to the release of CO2 from remineralization of particulate organic matter (see Wahyudi et al., 2021).

Additional comments

9.Line 321-333 and Conclusion: sampling coverage and chamber size (Fig 1-2) definitely a limitation of this study. I doubt whether the results of this research are sufficient to assess the carbon emission potential of all the studied seagrass areas. I suggest measuring the water- and air-pCO2 as well to see the potential carbon sink or source in the area. The authors need to clarify the term 'carbon emissions' whether as the release of CO2 into the water column, or from ecosystems into the air. This may help to explain, why the author choose mesocosm method compared to broadly measure the CO2 flux directly by measuring the carbonate system and pCO2.

Annotated reviews are not available for download in order to protect the identity of reviewers who chose to remain anonymous.

Reviewer 2 ·

Basic reporting

1. The English expression should be greatly improved over the MS. I suggest authors to find a fluent speaker to improve the English.
2. Over the abstract is just the results description. Abstract should only introduce the main results and the ecological implications.

Experimental design

Actually, authors have only measured the three times of the greenhouse gas flux in the seagrass meadows during 18 hours. Why authors only conducted 18 hours? It should be conducted at least 24 hours. Further, the authors measured the flux is positive, representing the emission from the seawater. This is not accurate due to the measuring method. Authors used the closed culture, which will make the greenhouse gas emission. I can accept used this method to measure the carbon metabolism but not the greenhouse gas emission. Actually, there is no previous studies have conducted the gas emission used this method.

Validity of the findings

1. In the discussion part, authors have compared the gas emission data with previous studies. Actually, they all used the different methods. Authors should not conduct this comparison.
2. Actually, most of the studies in the seagrass meadows, including the tropical seagrass meadows, have reported that the negative air-sea greenhouse gas emission. I suggest authors should firstly conduct the literature review.

Additional comments

No.

Reviewer 3 ·

Basic reporting

The authors have used clear and unambiguous English language throughout

Literature and references are well cited

The Figures and tables are professions

The results and hypothesis are well defined

Experimental design

Original primary research within Aims and Scope of the Journal

The research question is well defined is relevant and meaningful in Regional context and global context.

High technical and ethical standards are maintained

Methods explained with sufficient information for replication

Validity of the findings

Impact and novelity assessed

Data is robust and statistically sound

Conclusion is well stated

Annotated reviews are not available for download in order to protect the identity of reviewers who chose to remain anonymous.

---

## Round 0.2 · Minor Revisions

Reviewers suggested for minor revision, please follow their comments and revise accordingly. Authors can also check their title as per the reviewers comments in your next revision.

Reviewer 1 ·

Basic reporting

The manuscript presents valuable research findings on greenhouse gas emissions and carbon metabolism in tropical seagrass meadows. The authors have responded and revised the manuscript favorably. At this stage, I recommend it for publication after a minor revision.
I propose a title change to: "A Preliminary Study of Carbon dioxide and methane emissions from patchy tropical seagrass meadows in Thailand"

Experimental design

It improved well after revision.

Validity of the findings

As a preliminary study, the findings are interesting and valuable.

Additional comments

As mentioned, I propose a title change to: "A Preliminary Study of Carbon dioxide and methane emissions from patchy tropical seagrass meadows in Thailand"
The reasons are the limited scope of the study area, the limitations in the measurement method, and the limited aspects of the measurement (without measuring the difference in CO2 partial pressure at the study site).

Reviewer 2 ·

Basic reporting

I have one comment. Authors should mention they only conduct the CO2 and CH4 emission within 18 hours. They should mention this flaw in the discussion part.

Experimental design

Methods described with sufficient detail & information to replicate.

Validity of the findings

All underlying data have been provided; they are robust, statistically sound, & controlled.

Reviewer 3 ·

Basic reporting

No comment

Experimental design

No comment

Validity of the findings

No comment

Additional comments

Thanks to the authors for addressing all the previous comments/suggestions. However, I have found few more sentences requiring minor changes.

---

## Round 0.3 · accepted · Accept

The manuscript can be accepted based on all reviewers' comments. Congratulations to all authors.

Reviewer 1 ·

Basic reporting

The manuscript can be published.

Experimental design

The manuscript can be published.

Validity of the findings

The manuscript can be published.

Additional comments

The manuscript can be published.

Reviewer 3 ·

Basic reporting

The authors have addressed all of my concerns, and I am happy to recommend it for acceptance.

Experimental design

The experimental design is sound and well planned

Validity of the findings

The validity of the findings are important for the Indian Ocean region Seagrass Ecosystems